# Analysis of disease burden due to high body mass index in childhood asthma in China and the USA based on the Global Burden of Disease Study 2019

**Chengyue Zhang[1], Qing Qu[2], Kaiyu Pan**[ORCID][3]*

**1** Xiangya School of Medicine, Central South University, Changsha, Hunan, China, **2** Department of Nursing, The Second Affiliated Hospital of Zhejiang University, Hangzhou, Zhejiang, China, **3** Department of Pediatrics, The First People's Hospital of Xiaoshan District, Hangzhou, Zhejiang, China

* 804489145@qq.com

**Data Availability Statement:** All relevant files are available from the DANS EASY database: Zhang, C. Y. (Xiangya Hospital Central South University) (2022): Epidemiological data on childhood asthma

## Abstract

### Background

Currently, there is a growing concern about the disease burden of child asthma particularly due to high body mass index (BMI). The prevalence and disease burden of asthma differ between developing and developed countries, with implications on disease intervention. Therefore, we provide a comparative analysis of childhood asthma between China and the United States of America (USA).

### Methods

Using the Global Burden of Disease (GBD) 2019 data, we estimated and compared the age-standardized prevalence, disability-adjusted life years (DALYs), years of life lost (YLLs), years of lost due to disability (YLDs), DALYs due to high BMI of asthma in children aged 1–14 years in China and the USA. Joinpoint regression analysis was applied to assess changes in temporal trends.

### Results

DALYs due to high BMI and the ratio of DALYs to DALYs due to high BMI in children with asthma showed a significant upward trend in both countries and were higher in males than in females. Almost all epidemiological indicators of asthma showed a hump of curve from 2014 to 2019, and peaked in 2017. There was a decreasing trend of YLLs for asthma in children both countries, while China has a saliently greater decreasing trend.

### Conclusion

The disease burden caused by high BMI of childhood asthma was on the rise in children with asthma in both China and the USA. High BMI needs to be taken more into account in the development of future policies for the prevention, control, and treatment of childhood asthma. However, the increasing trend of this disease burden in American children was

in China and the United States. DANS. https://doi.
org/10.17026/dans-zvj-qgp4).

**Funding:** The author(s) received no specific
funding for this work.

**Competing interests:** The authors have declared
that no competing interests exist.

significantly lower than that in Chinese children. We recommend learning from the American government to impose a high-calorie tax, increase physical exercise facilities, and provide better health care policies.

## Introduction

Asthma is one of the most common chronic diseases in children, with wheezing, coughing, and airflow restriction as clinical manifestations, affecting children's daily life [1]. The incidence, prevalence, and medical costs of this disease have been increasing in recent years [2,3]. A survey revealed that the prevalence of asthma in Chinese children increased from 0.91% to 2.12% between 1990 and 2010 [4]. Respiratory health during early life may have a lifelong impact on lung health and life expectancy; thus, prevention and control of childhood asthma is particularly crucial to promote individual health and reduce the societal burden of the disease [5].

However, the etiology of childhood asthma is yet to be elucidated. Therefore, identifying its risk factors and exploring possible mechanisms is necessary for early detection and intervention to prevent further adverse outcomes [6]. Currently, reported risk factors for asthma include genetic factors, tobacco exposure, dampness/humidity, animal contact, climate, and inhalation of small particles [6–8]. High body mass index (BMI), which is considered as the seventh-leading level 2 risk factor for attributable disability-adjusted life years (DALYs) of diseases in 2019, is also a risk factor for asthma [9]. It is thought to be associated with dietary habits, lifestyle, and food intake [10].

There are differences in the prevalence and disease burden of asthma between developing and developed countries [3]. The direct and indirect economic costs of childhood asthma are high, and there is a link between the disease burden of asthma and the economic level of the country [11]. It is well known that China is the largest developing country in the world and that the United States of America (USA) is the major developed country [12,13]. However, to the best of our knowledge, there has been no comparative analysis between China and the USA in these areas. Thus, this study aimed to investigate the prevalence of asthma, DALYs, and the effect of the risk factor high BMI on disease burden in children aged 1–14 years in China and the USA, to compare and analyze the differences between them, to provide information for resource allocation, and to learn from the prevention and control strategies of developed countries such as the USA, which can provide some prevention and control strategies to reduce the disease burden of childhood asthma in developing countries such as China.

## Materials and methods

### Data source

The Global Burden of Disease (GBD) 2019 is a cross-border collaborative project covering 204 countries and regions. It collected data from disease surveillance sites, surveys of the National Health Service, and published literature data to estimate descriptive epidemiological information on the incidence, prevalence, disability-adjusted life years (DALYs), years of lost due to disability (YLDs), and years of life lost (YLLs) for 369 stratified diseases and injuries using the DisMod-MR 2.1 as a Bayesian meta-regression model [14,15]. The GBD estimation process uses 86,249 sources that are broad and representative, including censuses, household surveys,

health service use, civil registration and vital statistics, air pollution testing, etc. The data is publicly available (http://ghdx.healthdata.org/gbd-results-tool).

DALYs are a summary metric of YLDs calculated by multiplying the prevalence of individual sequelae by the disability weights, and YLLs that are the actual loss of life that occurs from death at each age modified by parameters such as the standard life expectancy at the corresponding age [16]. As a non-negligible risk factor for asthma, high BMI is defined as being overweight or obese by GBD 2019 for children aged 1–19 years according to International Obesity Task Force standards. In this study, we obtained data on the prevalence, DALYs, YLDs, YLLs, and DALYs due to high BMI of childhood asthma in children aged 1–14 years in China and the USA from GBD 2019. Furthermore, to evaluate the role played by high BMI in disease burden of childhood asthma, we calculated the ratio of DALYs to DALYs due to high BMI and plotted its temporal trend. To study the changes in each epidemiological indicator at different ages, we analyzed children aged 1–4 years, 5–9 years, and 10–14 years separately. All epidemiological data obtained were age-standardized to match the characteristics of the different national reference populations and finally expressed in terms of 100,000 population [14]. This study was conducted in compliance with the Guidelines for Accurate and Transparent Health Estimates Reporting (GATHER) [17]. No ethical review board approval was required for this study.

## Statistical analysis

Joinpoint Regression Program version 4.9.0.0 (National Cancer Institute, Rockville, MD, USA) was used to analyze the changes in trends in specific disease burden indicators from 1990 to 2019. The ordinary least squares method was used to fit the regression model under the assumption that the error random variables were homoskedastic. To increase the credibility of the results, we set the maximum number of joinpoints to 3 and determined the locations of the joinpoints and the corresponding p-values by Monte Carlo permutation test. Log-linear regression was used to calculate their annual percentage change (APC) and average annual percentage change (AAPC) [18]. T-distribution and normal distribution were used to assume sharp rocks for APC and AAPC, respectively, to evaluate whether the trend of curve changes and the overall trend of each segment are statistically significant, as well as to derive the 95% confidence intervals (CI). The AAPC is calculated from the APC in children from different countries, age groups and sex weighted geometrically by the length of each period. Comparing AAPC with 0, the curve shows an increasing or decreasing trend with 95% CI not including 0 when the AAPC value is positive or negative. When the 95% CI of AAPC includes 0, the value is stable. The differences were considered statistically significant at $P<0.05$.

## Results

### Trends in prevalence of childhood asthma in China and the USA from 1990 to 2019

The temporal trends in the age-standardized prevalence of childhood asthma by sex and age strata in China and the USA are presented in Fig 1, which can be further confirmed by the results of the joinpoint regression analysis (Table 1). Overall, the prevalence of childhood asthma showed a modest downward trend in Chinese children except for the group of male children aged 1–4 years, whereas it showed a modest upward trend in American children with higher disease prevalence. Compared with the previous change of decreasing and then increasing curve, the prevalence of asthma in children of all three age groups in China increased dramatically from 2014 to 2017, followed by a sharp decreasing trend starting from 2017. The prevalence of asthma among male and female children aged 1–4 years in the USA showed a

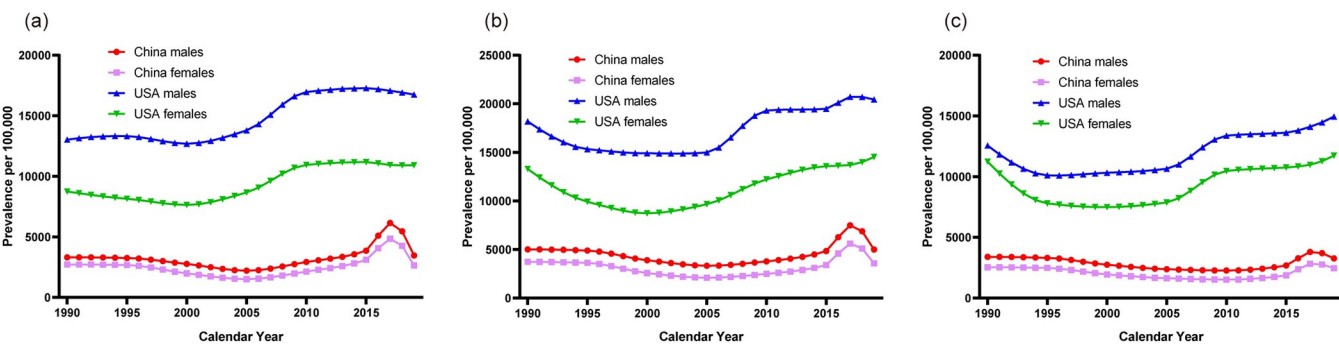

**Fig 1. Trends in the age-standardized prevalence per 100,000 population of children asthma by sex and age strata in China and the USA, 1990–2019.** (a). 1–4 years old age group. (b). 5–9 years old age group. (c). 10–14 years old age group.

relatively downward trend from 2010–2019 (males: AAPC -0.2; 95% CI -0.5, 0.1; females: AAPC -0.2; 95% CI -0.3, 0.0), and a small upward trend among children in the other two age groups. However, the AAPC for prevalence of asthma in the USA during this time period was significantly lower than that from 2005 to 2009, which means that the upward trend was reduced. In a cross-sectional comparison of the three age groups, the prevalence of asthma was higher in males than in females in all age groups, and the values of the prevalence were apparently higher in children aged 5–9 years than in the remaining two age groups both in China and the USA.

## Trends in DALYs and DALYs due to high BMI of childhood asthma in China and the USA from 1990 to 2019

The temporal trends in the age-standardized DALYs and DALYs due to high BMI of childhood asthma by sex and age strata in China and the USA are shown in Fig 2, while their APC and APCC values are detailed in Table 1. Temporal trends in DALYs and morbidity in childhood asthma were broadly similar, except for a large downward trend in Chinese children of both sexes aged 1–4 years (males: AAPC -1.3; 95% CI -2,8, 0.2; females: AAPC -2.2; 95% CI -4.1, -0.1). DALYs due to high BMI in children with asthma in the USA showed a consistent and substantial upward trend, and gradually decreased in 2010. The same indicator in Chinese children of almost all age group showed a hump of curve from 2014 to 2019, and peaked in 2017, except for that of Chinese females aged 10–14 years who showed a considerable increase from 2012 to 2019 (AAPC 13.0; 95% CI 10.5, 15.6). In terms of AAPC values, China is almost universally higher than the USA. Similarly, DALYs due to high BMI for asthma in children aged 5–9 years has the highest values among the three age groups and were higher in males than females in all age groups for both China and the USA.

## Trends in YLDs and YLLs of childhood asthma in China and the USA from 1990 to 2019

The temporal trends in the age-standardized YLDs and YLLs of childhood asthma by sex and age strata in China and the USA are shown in Fig 3 and supported with Table 2. The trend of YLDs was relatively identical to the prevalence of childhood asthma. Overall, there was a decreasing trend of YLLs for asthma in children both in China and the USA. The decreasing trend was saliently greater in China than in the USA, with the Chinese children aged 1–4 years having the highest onset of YLLs in 1990 and the largest absolute values of AAPC (males: AAPC -10.4; 95% CI -10.7, -10.0; females: AAPC -11.3; 95% CI -11.8, -10.8). Moreover, the

**Table 1. Trends in prevalence, DALYs rates and DALYs rates due to high BMI of children asthma by sex and age strata in China and the USA, 1990–2019, using Joinpoint regression models.**

| Measure | Age group | Sex | China | | | USA | | |
|---|---|---|---|---|---|---|---|---|
| | | | Time interval | APC (95% CI) | AAPC (95% CI) | Time interval | APC (95% CI) | AAPC (95% CI) |
| Prevalence | 1–4 years old | Male | 1990–2006 | -2.8 (-3.3, -2.3)* | 0.1 (-1.5, 1.7) | 1990–2002 | -0.3 (-0.5, -0.1)* | 0.8 (0.5, 1.2)* |
| | | | 2006–2014 | 5.6 (3.8, 7.5)* | | 2002–2005 | 2.5 (-0.8, 5.9) | |
| | | | 2014–2017 | 21.2 (6.6, 37.8)* | | 2005–2010 | 4.5 (3.5, 5.6)* | |
| | | | 2017–2019 | -23.2 (-32.5, -12.7)* | | 2010–2019 | -0.2 (-0.5, 0.1) | |
| | | Female | 1990–2006 | -4.2 (-4.8, -3.5)* | -0.3 (-2.4, 1.9) | 1990–2000 | -1.4 (-1.5, -1.3)* | 0.8 (0.6, 0.9)* |
| | | | 2006–2014 | 7.7 (5.2, 10.3)* | | 2000–2004 | 2.3 (1.5, 3.1)* | |
| | | | 2014–2017 | 21.4 (1.7, 45.0)* | | 2004–2010 | 5.0 (4.7, 5.4)* | |
| | | | 2017–2019 | -24.9 (-37.1, -10.3)* | | 2010–2019 | -0.2 (-0.3, 0.0)* | |
| | 5–9 years old | Male | 1990–2006 | -3.1 (-3.5, -2.7)* | -0.1 (-1.4, 1.2) | 1990–1994 | -4.1 (-5.0, -3.2)* | 0.4 (0.2, 0.7)* |
| | | | 2006–2014 | 3.6 (2.1, 5.0)* | | 1994–2005 | -0.4 (-0.6, -0.1)* | |
| | | | 2014–2017 | 20.5 (8.4, 34.0)* | | 2005–2009 | 6.3 (4.7, 8.0)* | |
| | | | 2017–2019 | -17.1 (-25.5, -7.9)* | | 2009–2019 | 0.9 (0.7, 1.2)* | |
| | | Female | 1990–2005 | -4.6 (-5.3, -4.0)* | -0.4 (-2.3, 1.6) | 1990–1995 | -5.9 (-6.6, -5.1)* | 0.3 (0.0, 0.5)* |
| | | | 2005–2014 | 4.0 (2.2, 5.8)* | | 1995–2001 | -2.3 (-3.1, -1.5)* | |
| | | | 2014–2017 | 24.9 (6.5, 46.5)* | | 2001–2012 | 3.9 (3.6, 4.2)* | |
| | | | 2017–2019 | -19.0 (-31.0, -5.0)* | | 2012–2019 | 1.4 (0.9, 1.9)* | |
| | 10–14 years old | Male | 1990–2008 | -2.6 (-2.9, -2.4)* | -0.3 (-1.4, 0.8) | 1990–1994 | -5.3 (-6.2, -4.3)* | 0.5 (0.2, 0.8)* |
| | | | 2008–2014 | 1.4 (-0.6, 3.4) | | 1994–2005 | 0.4 (0.1, 0.7)* | |
| | | | 2014–2017 | 16.2 (6.3, 27.1)* | | 2005–2009 | 5.5 (3.7, 7.2)* | |
| | | | 2017–2019 | -6.5 (-14.5, 2.3) | | 2009–2019 | 1.1 (0.8, 1.4)* | |
| | | Female | 1990–2008 | -3.3 (-3.7, -3.0)* | -0.3 (-1.8, 1.1) | 1990–1995 | -7.4 (-8.2, -6.5)* | 0.1 (-0.2, 0.4) |
| | | | 2008–2014 | 1.6 (-1.0, 4.2) | | 1995–2005 | 0.2 (-0.2, 0.5) | |
| | | | 2014–2017 | 19.7 (6.6, 34.4)* | | 2005–2009 | 7.4 (5.3, 9.5)* | |
| | | | 2017–2019 | -5.9 (-16.2, 5.6) | | 2009–2019 | 1.1 (0.8, 1.4)* | |
| DALYs | 1–4 years old | Male | 1990–2006 | -4.8 (-5.3, -4.4)* | -1.3 (-2.8, 0.2) | 1990–2001 | -0.4 (-0.6, -0.2)* | 0.8 (0.5, 1.1)* |
| | | | 2006–2014 | 4.7 (3.0, 6.5)* | | 2001–2005 | 1.8 (0.2, 3.4)* | |
| | | | 2014–2017 | 20.7 (6.5, 36.7)* | | 2005–2010 | 4.4 (3.4, 5.4)* | |
| | | | 2017–2019 | -23.0 (-32.1, -12.8)* | | 2010–2019 | -0.2 (-0.5, 0.1) | |
| | | Female | 1990–2005 | -7.4 (-8.0, -6.7)* | -2.2 (-4.1, -0.1) | 1990–2000 | -1.5 (-1.6, -1.3)* | 0.7 (0.6, 0.8)* |
| | | | 2005–2014 | 5.6 (3.7, 7.6)* | | 2000–2004 | 2.1 (1.3, 3.0)* | |
| | | | 2014–2017 | 22.0 (3.1, 44.4)* | | 2004–2010 | 4.8 (4.5, 5.2)* | |
| | | | 2017–2019 | -24.9 (-36.5, -11.1)* | | 2010–2019 | -0.2 (-0.4, -0.1)* | |
| | 5–9 years old | Male | 1990–2006 | -3.4 (-3.8, -3.1)* | -0.4 (-1.7, 0.9) | 1990–1994 | -3.9 (-4.8, -3.0)* | 0.4 (0.2, 0.7)* |
| | | | 2006–2014 | 3.3 (1.8, 4.8)* | | 1994–2005 | -0.4 (-0.6, -0.1)* | |
| | | | 2014–2017 | 20.1 (7.9, 33.6)* | | 2005–2009 | 6.2 (4.6, 7.8)* | |
| | | | 2017–2019 | -17.0 (-25.5, -7.7)* | | 2009–2019 | 0.9 (0.7, 1.1)* | |
| | | Female | 1990–2006 | -4.8 (-5.4, -4.1)* | -0.7 (-3.0, 1.6) | 1990–1995 | -5.6 (-6.4, -4.9)* | 0.3 (0.0, 0.5)* |
| | | | 2006–2014 | 4.4 (1.9, 7.0)* | | 1995–2001 | -2.3 (-3.0, -1.5)* | |
| | | | 2014–2017 | 23.8 (2.9, 48.9)* | | 2001–2012 | 3.8 (3.5, 4.1)* | |
| | | | 2017–2019 | -18.7 (-32.5, -2.2)* | | 2012–2019 | 1.4 (0.9, 1.9)* | |
| | 10–14 years old | Male | 1990–2007 | -3.1 (-3.4, -2.8)* | -0.7 (-1.8, 0.5) | 1990–1994 | -4.5 (-5.4, -3.6)* | 0.4 (0.2, 0.7)* |
| | | | 2007–2014 | 0.7 (-0.8, 2.2) | | 1994–2005 | 0.1 (-0.1, 0.3) | |
| | | | 2014–2017 | 15.8 (5.8, 26.6)* | | 2005–2009 | 4.9 (3.4, 6.5)* | |
| | | | 2017–2019 | -6.6 (-14.6, 2.1) | | 2009–2019 | 1.1 (0.8, 1.3)* | |
| | | Female | 1990–2008 | -3.8 (-4.3, -3.4)* | -0.8 (-2.6, 1.0) | 1990–1995 | -6.6 (-7.4, -5.8)* | 0.0 (-0.3, 0.4) |
| | | | 2008–2014 | 1.2 (-2.0, 4.6) | | 1995–2005 | -0.1 (-0.5, 0.2) | |
| | | | 2014–2017 | 18.9 (2.9, 37.5)* | | 2005–2009 | 6.9 (4.8, 8.9)* | |
| | | | 2017–2019 | -5.9 (-18.5, 8.8) | | 2009–2019 | 1.0 (0.7, 1.3)* | |

*(Continued)*

**Table 1.** (Continued)

| Measure | Age group | Sex | China | | | USA | | |
|---|---|---|---|---|---|---|---|---|
| | | | Time interval | APC (95% CI) | AAPC (95% CI) | Time interval | APC (95% CI) | AAPC (95% CI) |
| DALYs due to high BMI | 1–4 years old | Male | 1990–2006 | -0.4 (-1.0, 0.1) | 3.3 (1.5, 5.1)* | 1990–1993 | 3.1 (1.4, 4.8)* | 2.2 (1.9. 2.4)* |
| | | | 2006–2014 | 9.9 (7.8, 11.9)* | | 1993–2002 | 1.5 (1.1, 1.8)* | |
| | | | 2014–2017 | 24.3 (8.0, 43.0)* | | 2002–2010 | 4.4 (4.0, 4.9)* | |
| | | | 2017–2019 | -18.4 (-29.1, -6.2)* | | 2010–2019 | 0.6 (0.3, 0.9)* | |
| | | Female | 1990–2006 | -3.4 (-4.0, -2.8)* | 1.5 (-0.5, 3.6) | 1990–2000 | 0.6 (0.5, 0.7)* | 2.0 (1.8, 2.1)* |
| | | | 2006–2014 | 10.7 (8.3, 13.2)* | | 2000–2005 | 3.7 (3.2, 4.1)* | |
| | | | 2014–2017 | 24.5 (5.6, 46.8)* | | 2005–2010 | 5.6 (5.1, 6.1)* | |
| | | | 2017–2019 | -21.3 (-33.2, -7.2)* | | 2010–2019 | 0.5 (0.4, 0.6)* | |
| | 5–9 years old | Male | 1990–2006 | 1.2 (0.7, 1.7)* | 4.4 (2.9, 6.0)* | 1990–1994 | -0.6 (-1.6, 0.4) | 1.9 (1.5, 2.3)* |
| | | | 2006–2014 | 8.7 (7.0, 10.4)* | | 1994–2006 | 1.9 (1.7, 2.2)* | |
| | | | 2014–2017 | 24.1 (10.2, 39.7)* | | 2006–2009 | 7.2 (3.7, 10.8)* | |
| | | | 2017–2019 | -11.8 (-21.6, -0.7)* | | 2009–2019 | 1.3 (1.0, 1.6)* | |
| | | Female | 1990–2006 | -1.0 (-1.7, -0.3)* | 3.1 (0.7, 5.5)* | 1990–1995 | -3.3 (-4.0, -2.5)* | 1.7 (1.5, 2.0)* |
| | | | 2006–2014 | 8.3 (5.6, 11.0)* | | 1995–2001 | 0.3 (-0.5, 1.1) | |
| | | | 2014–2017 | 27.8 (6.1, 54.0)* | | 2001–2011 | 4.9 (4.6, 5.2)* | |
| | | | 2017–2019 | -15.2 (-29.6, 2.1) | | 2011–2019 | 2.1 (1.7, 2.5)* | |
| | 10–14 years old | Male | 1990–2008 | 1.9 (1.5, 2.3)* | 4.3 (2.7, 6.0)* | 1990–1994 | -1.8 (-2.7, -1.0)* | 1.8 (1.5, 2.1)* |
| | | | 2008–2014 | 6.3 (3.3, 9.3)* | | 1994–2006 | 2.2 (2.0, 2.4)* | |
| | | | 2014–2017 | 19.7 (5.6, 35.8)* | | 2006–2009 | 6.4 (3.5, 9.3)* | |
| | | | 2017–2019 | -0.6 (-12.4, 12.7) | | 2009–2019 | 1.4 (1.2, 1.6)* | |
| | | Female | 1990–1996 | 4.0 (1.1, 7.1)* | 4.1 (2.9, 5.3)* | 1990–1994 | -5.3 (-6.3, -4.3)* | 1.3 (1.0, 1.5)* |
| | | | 1996–2005 | -1.7 (-3.5, 0.2) | | 1994–2005 | 1.6 (1.3, 1.8)* | |
| | | | 2005–2012 | 3.1 (0.2, 6.1)* | | 2005–2009 | 7.5 (5.8, 9.3)* | |
| | | | 2012–2019 | 13.0 (10.5, 15.6)* | | 2009–2019 | 1.2 (1.0, 1.5)* | |

* Significantly different from 0 (P < 0.05).

*Abbreviation*: DALYs, disability-adjusted life years; BMI, body mass index; APC, annual percent change; AAPC, average annual percent change; CI, confidential interval.

YLLs for children aged 10–14 years in the USA have the highest levels than the other groups. YLLs were also higher in males than that in females in all age groups of children.

### Trends in ratio of DALYs rates to DALYs rates due to high BMI of childhood asthma in China and the USA from 1990 to 2019

The ratio of DALYs to DALYs due to high BMI in children with asthma showed a consistent upward trend in both China and the USA (Table 3 and Fig 4). The ratio was higher in the USA than in China, and the AAPC in China was approximately 3–4 times higher than that in the USA, suggesting that DALYs due to high BMI of asthma had a higher increasing trend in Chinese children. In terms of age group, Chinese children aged 10–14 years and American children aged 5–9 years had the highest increasing trend of this ratio respectively. Notably, the increasing trend of the ratio is apparently higher in male children than in female children in China, while there was no obvious difference between the two sexes in the USA.

### Discussion

Our findings revealed that the prevalence of asthma among children of all ages in China showed a slow increase from around 2006 to 2014, which can be attributed with the

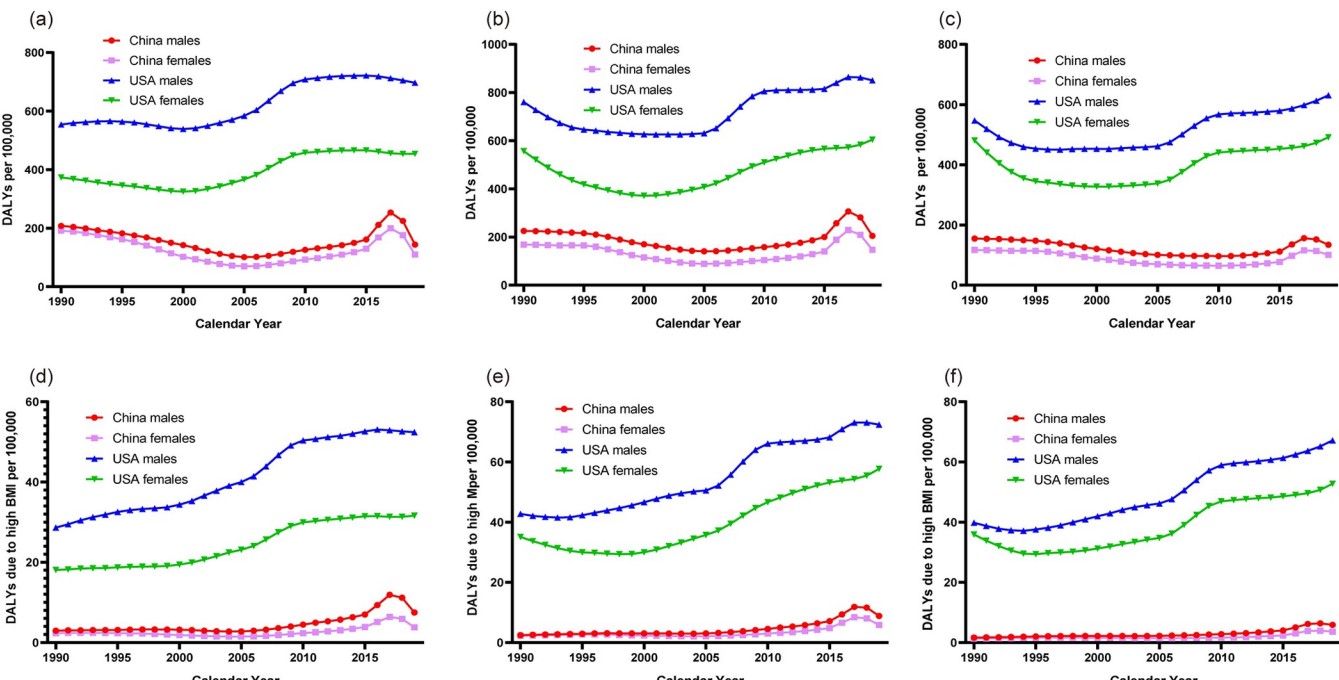

**Fig 2. Trends in the age-standardized DALYs rates and DALYs rates due to high BMI per 100,000 population of children asthma by sex and age strata in China and the USA, 1990–2019.** (a). DALYs in 1–4 years old age group. (b). DALYs in 5–9 years old age group. (c). DALYs in 10–14 years old age group. (d). DALYs due to high BMI in 1–4 years old age group. (e). DALYs due to high BMI in 5–9 years old age group. (f). DALYs due to high BMI in 10–14 years old age group. Abbreviation: DALYs, disability-adjusted life years; BMI, body mass index.

environmental pollution due to industrialization and urbanization in China, as well as the increase in children with high BMI [19,20]. The rapid increase of prevalence from 2016 to 2017 is related to the release of updated guidelines on the diagnosis and prevention of childhood asthma in China in 2016. In this guideline, clear quantitative indicators for the diagnosis of asthma in children are proposed, thus effectively improving the level of diagnosis [21]. The rapid decline from 2017 to 2019 may be the result of the first Chinese Children's Asthma Action Plan (CCAAP) released in China in 2017, which combines doctors' treatment decisions, education on basic asthma treatment and children's compliance with medical advice, providing a standardized and personalized basis for child treatment and family management, thus enhancing asthma control [22]. DALYs of asthma in children of all ages in China were decreasing, which could be attributed to the release of asthma guidelines and the promotion of formal treatment protocols. These strategies can significantly reduce the disease burden by reducing disease severity and improving symptom control [21,23,24].

Overall, the prevalence and DALYs of childhood asthma in the USA showed an increasing trend from 1990 to 2019. Part of the reason for this is related to the significant increase in the consumption of sugary drinks such as nutritional/energy drinks, juice drinks, and sweet tea among American children, and sugary drinks are thought to be associated with childhood asthma [25]. However, the upward trend of both indicators from around 2010 to 2019 was significantly reduced compared to that in the period from 2005 to 2009. The prevalence and DALYs of childhood asthma decreased in children aged 1–4 years between 2010 and 2019. This may be related to the enactment of the Clean Air Act amendments in 2011 and the Clean Power Plan in 2015 in the USA. In particular, the Regional Greenhouse Gas Initiative (RGGI) has contributed significantly to the reduction of greenhouse gases in the power sector and

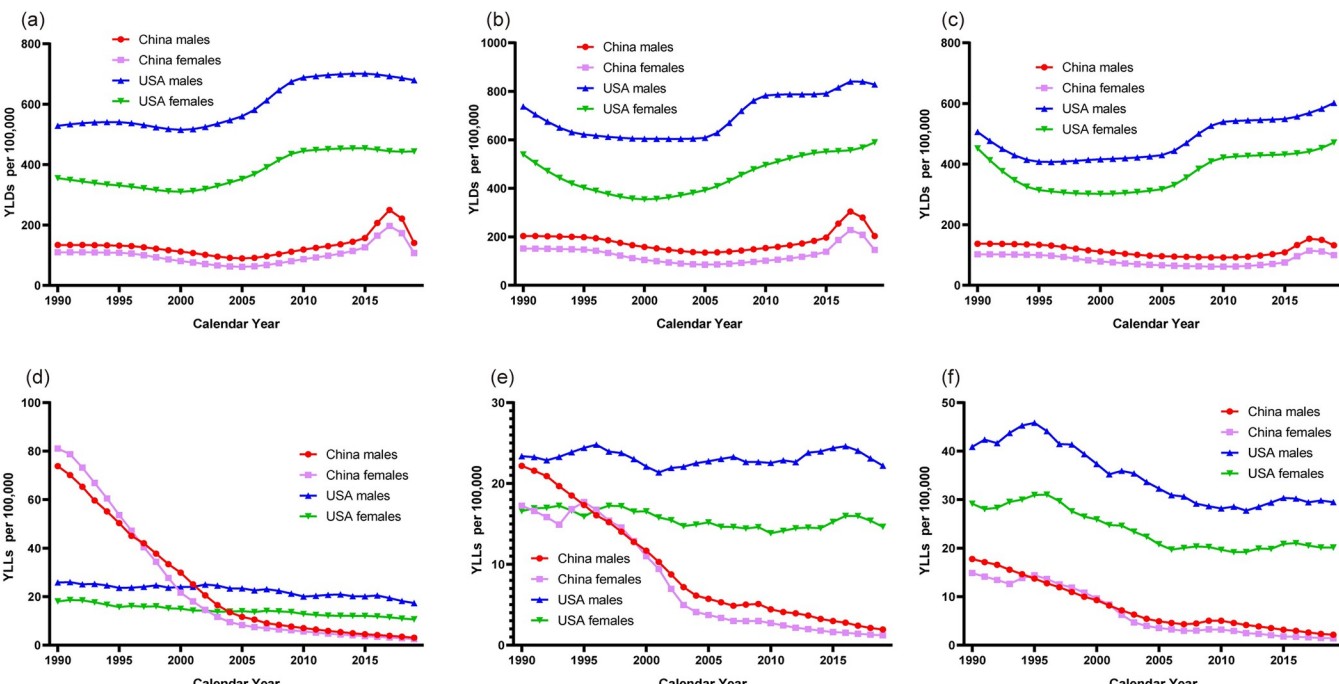

**Fig 3. Trends in the age-standardized YLDs rates and YLLs rates due to high BMI per 100,000 population of children asthma by sex and age strata in China and the USA, 1990–2019.** (a). YLDs in 1–4 years old age group. (b). YLDs in 5–9 years old age group. (c). YLDs in 10–14 years old age group. (d). YLLs in 1–4 years old age group. (e). YLLs in 5–9 years old age group. (f). YLLs in 10–14 years old age group. Abbreviation: YLDs, years lived with disability; YLLs, years of life lost.

toxic air pollutants associated with the onset of asthma, such as PM2.5, over the past decade [26]. Another possible reason is that most US states enacted a tax on sugar-sweetened beverages (SSB) by 2010, which plays a role in reducing SSB-induced high BMI [27]. The prevalence and DALYs of asthma were higher in children in the USA than those in China. The main reason may be related to the health hypothesis, indicating that less exposure to infection during childhood may result in a greater chance of developing asthma later in life [3]. YLLs in children in both countries account for a minimal proportion of DALYs, and are associated with a very low mortality rate in children with asthma [28].

High BMI is a risk factor of great concern for childhood asthma [29]. Our study found an increasing trend in DALYs due to high BMI and the ratio of DALYs rates to DALYs rates due to high BMI in Chinese and American children with asthma. One reason for this is that certain parents of children with asthma are concerned about exercise-induced bronchoconstriction (EIB), and thus impose restrictions on the physical activities of their children, consequently affecting weight management [30]. However, in reality, exercise can increase lung function, promote cardiopulmonary fitness, and control asthma [31]. Adequate warm-up before exercise is also recommended [32]. Another reason is that children with high BMI are more inclined to consume a high-fat diet, which can increase bronchial hyperresponsiveness and exacerbate the symptoms of asthma [33]. Positive effects of weight loss on asthma-related outcomes have been demonstrated [34]. The latest Global Initiative for Asthma (GINA 2021) lists obesity as a modifiable risk factor [29]. Since the global prevalence of high BMI is constantly rising, increasing exercise and reducing high-fat food intake to control high BMI are necessary to reduce the risk of asthma [35]. Although the DALYs rates due to high BMI and the ratio of DALYs rates to DALYs rates due to high BMI were both higher in American children than in

**Table 2. Trends in YLDs rates and YLLs rates by sex and age strata in China and the USA, 1990–2019, using Joinpoint regression models.**

| Measure | Age group | Sex | China | | | USA | | |
|---|---|---|---|---|---|---|---|---|
| | | | Time interval | APC (95% CI) | AAPC (95% CI) | Time interval | APC (95% CI) | AAPC (95% CI) |
| YLDs | 1–4 years old | Male | 1990–2006 | -2.8 (-3.3, -2.3)* | 0.1 (-1.5, 1.7) | 1990–2001 | -0.4 (-0.6, -0.2)* | 0.8 (0.5, 1.1)* |
| | | | 2006–2014 | 5.6 (3.8, 7.5)* | | 2001–2005 | 1.9 (0.2, 3.6)* | |
| | | | 2014–2017 | 21.2 (6.6, 37.7)* | | 2005–2010 | 4.6 (3.5, 5.7)* | |
| | | | 2017–2019 | -23.2 (-32.5, -12.7)* | | 2010–2019 | -0.2 (-0.5, 0.1) | |
| | | Female | 1990–2006 | -4.2 (-4.8, -3.5)* | -0.3 (-2.4, 1.9) | 1990–2000 | -1.4 (-1.5, -1.3)* | 0.8 (0.7, 0.9)* |
| | | | 2006–2014 | 7.7 (5.2, 10.3)* | | 2000–2004 | 2.3 (1.5, 3.1)* | |
| | | | 2014–2017 | 21.4 (1.7, 44.9)* | | 2004–2010 | 5.0 (4.7, 5.4)* | |
| | | | 2017–2019 | -24.8 (-37.0, -10.3)* | | 2010–2019 | -0.2 (-0.3, 0.0)* | |
| | 5–9 years old | Male | 1990–2006 | -3.0 (-3.4, -2.7)* | -0.1 (-1.4, 1.2) | 1990–1994 | -4.1 (-5.0, -3.2)* | 0.4 (0.2, 0.7)* |
| | | | 2006–2014 | 3.6 (2.1, 5.0)* | | 1994–2005 | -0.4 (-0.6, -0.1)* | |
| | | | 2014–2017 | 20.5 (8.4, 33.9)* | | 2005–2009 | 6.3 (4.7, 8.0)* | |
| | | | 2017–2019 | -17.1 (-25.4, -7.9)* | | 2009–2019 | 0.9 (0.7, 1.2)* | |
| | | Female | 1990–2005 | -4.6 (-5.3, -4.0)* | -0.4 (-2.3, 1.6) | 1990–1995 | -5.8 (-6.6, -5.1)* | 0.3 (0.0, 0.5)* |
| | | | 2005–2014 | 4.0 (2.2, 5.8)* | | 1995–2001 | -2.3 (-3.1, -1.5)* | |
| | | | 2014–2017 | 24.9 (6.6, 46.5)* | | 2001–2012 | 3.9 (3.6, 4.2)* | |
| | | | 2017–2019 | -19.0 (-30.9, -5.1)* | | 2012–2019 | 1.4 (0.9, 1.9)* | |
| | 10–14 years old | Male | 1990–2008 | -2.6 (-2.9, -2.3)* | -0.3 (-1.4, 0.8) | 1990–1994 | -5.3 (-6.2, -4.3)* | 0.5 (0.2, 0.8)* |
| | | | 2008–2014 | 1.4 (-0.6, 3.4) | | 1994–2005 | 0.4 (0.2, 0.7)* | |
| | | | 2014–2017 | 16.2 (6.3, 27.1)* | | 2005–2009 | 5.5 (3.8, 7.2)* | |
| | | | 2017–2019 | -6.5 (-14.5, 2.2) | | 2009–2019 | 1.1 (0.8, 1.3)* | |
| | | Female | 1990–2008 | -3.3 (-3.7, -2.9)* | -0.3 (-1.8, 1.1) | 1990–1995 | -7.3 (-8.1, -6.5)* | 0.1 (-0.2, 0.4) |
| | | | 2008–2014 | 1.6 (-1.0, 4.2) | | 1995–2005 | 0.2 (-0.2, 0.5) | |
| | | | 2014–2017 | 19.7 (6.6, 34.4)* | | 2005–2009 | 7.4 (5.3, 9.5)* | |
| | | | 2017–2019 | -5.9 (-16.2, 5.6) | | 2009–2019 | 1.1 (0.8, 1.4)* | |
| YLLs | 1–4 years old | Male | 1990–1994 | -7.1 (-8.5, -5.6)* | -10.4 (-10.7, -10.0)* | 1990–2007 | -0.5 (-0.8, -0.3)* | -1.3 (-2.1, -0.5)* |
| | | | 1994–2000 | -9.6 (-10.6, -8.6)* | | 2007–2010 | -4.1 (-10.7, 3.0) | |
| | | | 2000–2005 | -18.2 (-19.5, -16.9)* | | 2010–2016 | 0.0 (-1.6, 1.6) | |
| | | | 2005–2019 | -8.7 (-8.9, -8.5)* | | 2016–2019 | -5.4 (-8.7, -1.9)* | |
| | | Female | 1990–1994 | -7.0 (-8.8, -5.2)* | -11.3 (-11.8, -10.8)* | 1990–2003 | -2.2 (-2.6, -1.9)* | -1.8 (-2.6, -1.0)* |
| | | | 1994–1998 | -13.6 (-16.2, -10.9)* | | 2003–2008 | 0.2 (-2.1, 2.5) | |
| | | | 1998–2004 | -19.8 (-20.9, -18.7)* | | 2008–2011 | -3.4 (-10.2, 4.0) | |
| | | | 2004–2019 | -8.1 (-8.3, -7.9)* | | 2011–2019 | -1.8 (-2.5, -1.0)* | |
| | 5–9 years old | Male | 1990–2000 | -6.3 (-7.0, -5.5)* | -8.2 (-8.9, 7.5)* | 1990–1996 | 1.2 (0.3, 2.2)* | -0.1 (-0.6, 0.4) |
| | | | 2000–2004 | -16.2 (-20.2, -12.1)* | | 1996–2001 | -2.4 (-4.1, -0.7)* | |
| | | | 2004–2012 | -4.9 (-6.2, -3.7)* | | 2001–2017 | 0.7 (0.4, 0.9)* | |
| | | | 2012–2019 | -9.8 (-10.9, -8.6)* | | 2017–2019 | -4.4 (-9.6, 1.0) | |
| | | Female | 1990–1996 | 0.6 (-1.8, 3.1) | -8.6 (-9.9, -7.3)* | 1990–1998 | -0.0 (-0.8, 0.7) | -0.5 (-1.1, 0.1) |
| | | | 1996–2000 | -8.9 (-15.3, -2.1)* | | 1998–2010 | -1.6 (-2.1, -1.1)* | |
| | | | 2000–2004 | -23.0 (-28.3, -17.2)* | | 2010–2017 | 2.0 (0.8, 3.3)* | |
| | | | 2004–2019 | -7.8 (-8.4, -7.2)* | | 2017–2019 | -4.0 (-10.6, 3.2) | |
| | 10–14 years old | Male | 1990–1997 | -5.4 (-6.2, -4.6)* | -7.1 (-7.8, -6.5)* | 1990–1995 | 1.9 (0.6, 3.1)* | -1.1 (-1.6, -0.7)* |
| | | | 1997–2007 | -10.3 (-10.8, -9.8)* | | 1995–2010 | -3.2 (-3.5, -3.0)* | |
| | | | 2007–2010 | 7.5 (0.8, 14,5)* | | 2010–2015 | 1.7 (-0.0, 3.5) | |
| | | | 2010–2019 | -9.3 (-9.8, -8.8)* | | 2015–2019 | -0.4 (-2.1, 1.4) | |
| | | Female | 1990–1999 | -2.3 (-3.7, -0.9)* | -8.0 (-9.0, -6.9)* | 1990–1996 | 1.4 (0.2, 2.6)* | -1.1 (-1.6, 0.6)* |
| | | | 1999–2005 | -19.4 (-22.1, -16.6)* | | 1996–2006 | -4.1 (-4.7, -3.5)* | |
| | | | 2005–2010 | -1.3 (-5.9, 3.6) | | 2006–2011 | -0.7 (-2.8, 1.5) | |
| | | | 2010–2019 | -8.9 (-10.2, -7.6)* | | 2011–2019 | 0.7 (-0.1, 1.4) | |

* Significantly different from 0 (P < 0.05).

*Abbreviation*: YLDs, years lived with disability; YLLs, years of life lost; APC, annual percent change; AAPC, average annual percent change; CI, confidential interval.

**Table 3. Trends in ratio of DALYs rates to DALYs rates due to high BMI of children asthma by sex and age strata in China and the USA, 1990–2019, using Joinpoint regression models.**

| Measure | Age group | Sex | China | | | USA | | |
|---|---|---|---|---|---|---|---|---|
| | | | Time interval | APC (95% CI) | AAPC (95% CI) | Time interval | APC (95% CI) | AAPC (95% CI) |
| ratio of DALYs rates to DALYs rates due to high BMI | 1–4 years old | Male | 1990–2008 | 4.6 (4.5, 4.7)* | 4.6 (4.2, 5.1)* | 1990–1992 | 2.5 (1.4, 3.5)* | 1.3 (1.2. 1.4)* |
| | | | 2008–2011 | 6.0 (2.4, 9.8)* | | 1992–2003 | 2.1 (2.0, 2.1)* | |
| | | | 2011–2016 | 3.5 (2.3, 4.6)* | | 2003–2014 | 0.6 (0.5, 0.7)* | |
| | | | 2016–2019 | 5.2 (3.4, 7.1)* | | 2014–2019 | 0.8 (0.6, 1.1)* | |
| | | Female | 1990–1995 | 3.4 (3.0, 3.7)* | 3.6 (3.5. 3.8)* | 1990–1993 | 2.3 (1.9, 2.1)* | 1.3 (1.2, 1.3)* |
| | | | 1995–2000 | 4.7 (4.1, 5.2)* | | 1993–2003 | 2.0 (1.9, 2.1)* | |
| | | | 2000–2005 | 3.1 (2.5, 3.6)* | | 2003–2006 | -0.1 (-0.8, 0.7) | |
| | | | 2005–2019 | 3.6 (3.5, 3.6)* | | 2006–2019 | 0.8 (0.7, 0.8)* | |
| | 5–9 years old | Male | 1990–2000 | 5.3 (4.9, 5.5)* | 4.9 (4.6, 5.1)* | 1990–1995 | 3.1 (2.9, 3.2)* | 1.4 (1.4, 1.5)* |
| | | | 2000–2005 | 3.9 (2.8, 5.0)* | | 1995–2000 | 2.6 (2.4, 2.9)* | |
| | | | 2005–2011 | 5.8 (5.1, 6.6)* | | 200–2003 | 2.1 (1.4, 2.8)* | |
| | | | 2011–2019 | 4.4 (4.0, 4,7)* | | 2003–2019 | 0.4 (0.4, 0.5)* | |
| | | Female | 1990–1995 | 3.6 (3.2, 4.1)* | 3.8 (3.6, 4.0)* | 1990–2003 | 2.5 (2.4, 2.5)* | 1.4 (1.4, 1.5)* |
| | | | 1995–1999 | 4.9 (3.9, 6.0)* | | 2003–2006 | 0.5 (-0.0, 1.0) | |
| | | | 1999–2005 | 3.4 (2.9, 3.9)* | | 2006–2010 | 1.0 (0.7, 1.3)* | |
| | | | 2005–2019 | 3.8 (3.7, 3.9)* | | 2010–2019 | 0.5 (0.4, 0.5)* | |
| | 10–14 years old | Male | 1990–2000 | 5.8 (4.8, 5.3)* | 5.1 (1.5, 5.1)* | 1990–1994 | 2.7 (2.5, 2.9)* | 1.3 (1.3, 1.4)* |
| | | | 2000–2005 | 3.7 (2.8, 4.7)* | | 1994–2003 | 2.2 (2.1, 2.3)* | |
| | | | 2005–2011 | 5.6 (4.9, 6.2)* | | 2003–2012 | 0.7 (0.6, 0.8)* | |
| | | | 2011–2019 | 4.5 (4.2, 4.9)* | | 2012–2019 | 0.2 (0.2, 0.3)* | |
| | | Female | 1990–1995 | 4.4 (4.0, 4.8)* | 4.1 (3.9. 4.2)* | 1990–1994 | 2.8 (2.7, 2.9)* | 1.3 (1.2, 1.3)* |
| | | | 1995–1999 | 5.3 (4.4, 6.2)* | | 1994–2003 | 2.2 (2.2, 2.2)* | |
| | | | 1999–2004 | 3.4 (2.8, 3.9)* | | 2003–2011 | 0.6 (0.6, 0.7)* | |
| | | | 2004–2019 | 3.9 (3.8, 3.9)* | | 2011–2019 | 0.1 (0.0, 0.1)* | |

* Significantly different from 0 (P < 0.05).

*Abbreviation*: DALYs, disability-adjusted life years; BMI, body mass index; APC, annual percent change; AAPC, average annual percent change; CI, confidential interval.

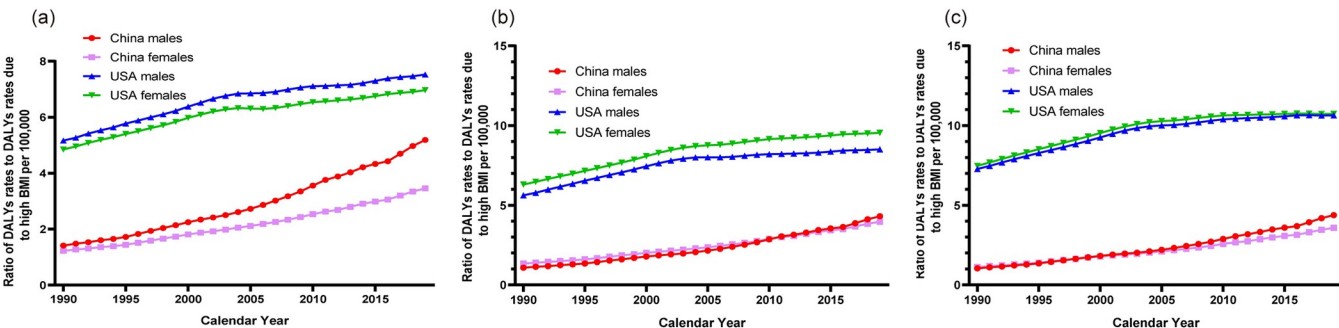

**Fig 4. Trends in the in the ratio of DALYs due to high BMI/DALYs of children asthma by sex and age strata in China and the USA, 1990–2019.** (a). 1–4 years old age group. (b). 5–9 years old age group. (c). 10–14 years old age group. Abbreviation: DALYs, disability-adjusted life years; BMI, body mass index.

Chinese children, the increasing trend of both indicators was significantly higher in Chinese children of almost all ages than in the American children. Chinese boys of all ages had a higher percentage of high BMI than girls, which is consistent with the findings of Guo et al. [36]. A study of disease burden, injury, and risk factors by state in the USA from 1990 to 2016 found that high BMI was the most important risk factor in the USA, and that exposure was steadily increasing [37]. They believe that renewed efforts to control weight at the community level are important, and that controlling high BMI needs to be a priority for all stakeholders such as physicians, nurses, policy makers, patients, and families. Liu et al. found that the American government partially eliminated the adverse effects of obesity on asthma by imposing a high-calorie tax, increasing the proportion of nutritious food advertisements, banning the sale of soft drinks, increasing opportunities and venues for physical activity, and implementing better health care policies [38]. Therefore, it is recommended that high BMI be taken more into account in the future development of policies for the prevention, control, and treatment of childhood asthma. Moreover, we recommend that children reduce their BMI by increasing physical activity and eating a healthy diet, which parents should encourage and safeguard [39].

The prevalence, DALYs and DALYs due to high BMI were higher in boys than in girls across all age groups in both countries. Boys also have a relatively narrow airway, and are more inclined to vigorous exercise with their greater range of motion, and thus, are more likely to get exposed to allergens. This corroborates the findings of Ellie et al. that boys are more likely to develop allergic diseases than girls based on blood-specific IgE assays and skin prick tests for common allergens [7]. In both China and the USA, the prevalence and DALYs were highest in the 5–9 years age group. This could be due to the fact that children in this age group are at high risk of upper respiratory tract infections because of their low self-management skills and immune levels [36]. In addition, upper respiratory tract infections are important triggers for asthma in children. Moreover, the YLLs for girls aged 10–14 years in the USA have the highest levels than that for girls in the other groups, which is considered to be associated with the increased levels of estrogen and progesterone during the luteal phase in girls of this age group, resulting in increased inflammation of the airway wall [40].

Our study has some limitations. The GBD 2019 lacks data on other risk factors such as high-fat diet and tobacco exposure for asthma in children aged 1–14 years, as well as interactions between risk factors in the estimates, these factors may have introduced bias in the study. In addition, the diagnosis of asthma in children aged 1–4 years is based primarily on clinical judgment and assessment of symptoms and physical findings, which may lead to a failure to reliably diagnose asthma in this age group [4,41].

## Conclusion

DALYs rates due to high BMI and ratio of DALYs rates to DALYs rates due to high BMI were on the rise in children with asthma in both China and USA. High BMI needs to be taken more into account in the development of future policies for the prevention, control, and treatment of childhood asthma. Although both indicators of asthma in children in the USA are higher than in those China, the increasing trend is significantly lower than that in Chinese children of almost all ages. Therefore, it is recommended to learn from the American government to impose a high-calorie tax, increase physical exercise facilities, and provide better health care policies. Besides, we appeal that children increase their physical activity and maintain a healthy diet and that parents encourage and safeguard it. Children aged 5–9 years had the highest prevalence and DALYs in both countries. Careful attention and targeted intervention should be considered in this population, who are particularly at high risk for asthma.

## Acknowledgments

The authors would like to thank Editage (www.editage.cn) for English language editing.

## Author Contributions

**Conceptualization:** Chengyue Zhang, Kaiyu Pan.

**Data curation:** Chengyue Zhang, Qing Qu.

**Methodology:** Chengyue Zhang.

**Project administration:** Kaiyu Pan.

**Software:** Qing Qu.

**Writing – original draft:** Chengyue Zhang, Qing Qu.

**Writing – review & editing:** Kaiyu Pan.

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
