## [Decision Letter · Decision Letter 0]

4 Jan 2023

PONE-D-22-30916Analysis of disease burden due to high body mass index in childhood asthma in China and the USA based on the Global Burden of Disease Study 2019PLOS ONE

Dear Dr. Pan,

Thank you for submitting your manuscript to PLOS ONE. After careful consideration, we feel that it has merit but does not fully meet PLOS ONE’s publication criteria as it currently stands. Therefore, we invite you to submit a revised version of the manuscript that addresses the points raised during the review process.

ACADEMIC EDITOR: Dear authors, please revise your manuscript according to the referees’ and editor's comments and upload the revised file. First thing is that you need to revise rationale for a comparison of China and USA. You just mentioned that there is no such a study and that you decided to do it, however, you have to state precisely why is it so important, to make a comparison exactly between these two countries. Furthermore, the aims in your Introduction section are mentioned twice and are not completely clear. Be sure to define them accurately. Both reviewers indicated that the manuscript lacks in more details on situation in USA. Further, try to provide more in-depth analysis throughout the whole manuscript. Finally, please, double-check the reference number 22.

We look forward to receiving your revised manuscript.

Kind regards,

Gorica Maric

Academic Editor

PLOS ONE

Journal Requirements:

Reviewers' comments:

Reviewer's Responses to Questions

**Comments to the Author**

1. Is the manuscript technically sound, and do the data support the conclusions?

Reviewer #1: Partly

Reviewer #2: Partly

2. Has the statistical analysis been performed appropriately and rigorously? 

Reviewer #1: Yes

Reviewer #2: Yes

3. Have the authors made all data underlying the findings in their manuscript fully available?

Reviewer #1: Yes

Reviewer #2: Yes

4. Is the manuscript presented in an intelligible fashion and written in standard English?

Reviewer #1: Yes

Reviewer #2: Yes

5. Review Comments to the Author

Reviewer #1: Dear authors:

We are very grateful for your submission of the manuscript to Plos One. The research topic is considered to be very interesting but the article requires an in-depth review of the results and, in particular, a further discussion of childhood asthma in the USA. Also, the introduction, materials and methods are not detailed enough.

Sincerely.

Reviewer #2: The authors have chosen an interesting study topic to quantify the disease burden due to high body mass index in childhood asthma. The GDB methodology is currently a very important approach to global descriptive epidemiology with the intention of knowledge translation. Hence, the results of the GBD studies should be understandable not only to the academic audience but also to the stakeholders and decision-makers in the health policy domain. In this sense, all improvements to the text should enable the achievement of these goals. There are no ethical issues concerning this paper.

The text in the Introduction (lines 57-59) needs a reference. Consider excluding Ref 8 and deleting the sentence from P 4, line 60. Also, the authors should include an additional reference which is recently published on this topic (Jin Liu, Maobo Yuan, Yuqian Chen, Yan Wang, Qingting Wang, Qianqian Zhang, Limin Chai, Danyang Li, Yuanjie Qiu, Huan Chen, Jian Wang, Xinming Xie, Manxiang Li. Global burden of asthma associated with high body mass index from 1990 to 2019, Annals of Allergy, Asthma & Immunology 2022; 129:6, 720-730)

The Protocol for the global burden of diseases, injuries, and risk factors study (Institute for Health Metrics and Evaluation, 2018) should be the reference added (and used) to the Method section.

The results are extensive. However, it seems difficult for authors to extract a meaningful message from the results section to emphasize the rationale for comparing the findings from China and USA. They haven’t concluded anything about disease burden due to the high body mass index in childhood asthma in the USA (P 29, lines 261-269). Thus, I suggest rewriting of the conclusion section.

In the Discussion section, I suggest that authors delete the sentence (P 22, lines 237-239); it is irrelevant to this paper.

I am confused with the text in the Reference section (lines 360-457). Please, check if it is an addition of ref. 22, and, is it necessary.

6. PLOS authors have the option to publish the peer review history of their article (what does this mean?). If published, this will include your full peer review and any attached files.

Reviewer #1: **Yes: **Enrique Gea-Izquierdo

Reviewer #2: No

---

## [Author Response · Author response to Decision Letter 0]

2 Feb 2023

February 2, 2023

Dear Editor,

Thank you very much for giving us the opportunity to revise our manuscript entitled “Analysis of disease burden due to high body mass index in childhood asthma in China and the USA based on the Global Burden of Disease Study 2019”. The manuscript ID is PONE-D-22-30916. We would also like to thank the editorial board and all the reviewers for their constructive and insightful comments, which helped us improve our manuscript.

We have ensured that all reviewer comments were adequately answered. For example, in accordance with the suggestions given by Reviewer 1, we have enriched the content of the “Introduction” and “Materials and methods” sections. As for the concerns raised by the two reviewers and the editor, we have added a more in-depth discussion of the prevalence, DALYs, and effect of the risk factor high BMI on disease burden of childhood asthma in children in the USA. Responses to the reviewers’ comments are provided below in a point-by-point manner. All the changes made are marked in red in the revised manuscript. We hope that our revisions satisfactorily address all concerns raised by the reviewers.

In addition, we reworked the files to confirm compliance with journal style requirements. For example, our figure files have been corrected by the Preflight Analysis and Conversion Engine (PACE) digital diagnostic tool. Besides, we have uploaded the data from this study to a stable public repository. The data links are as follows: Zhang, C.Y. (Xiangya Hospital Central South University) (2022): Epidemiological data on childhood asthma in China and the United States. DANS. https://doi.org/10.17026/dans-zvj-qgp4. 

We wish to thank you again for carefully reviewing and considering our manuscript. 

Sincerely yours,

Chengyue Zhang

Response to Comments

Academic Editor

Dear authors, please revise your manuscript according to the referees’ and editor's comments and upload the revised file. First thing is that you need to revise rationale for a comparison of China and USA. You just mentioned that there is no such a study and that you decided to do it, however, you have to state precisely why is it so important, to make a comparison exactly between these two countries. Furthermore, the aims in your Introduction section are mentioned twice and are not completely clear. Be sure to define them accurately. Both reviewers indicated that the manuscript lacks in more details on situation in USA. Further, try to provide more in-depth analysis throughout the whole manuscript. Finally, please, double-check the reference number 22.

Response: Thank you for this comment. We have added a rationale for choosing China and the USA for comparison, as well as a more in-depth discussion of the prevalence, DALYs, and effect of the risk factor high BMI on disease burden of childhood asthma in children in the USA. In addition, we apologize for not clearly expressing the objectives in the “Introduction” section; thus, we have reordered the statements, expanded the content, and optimized the “Introduction” section. For reference number 22, we apologize for this error. We have checked the style of the cited reference and have made corrections.

Reviewer 1

Dear authors:

We are very grateful for your submission of the manuscript to Plos One. The research topic is considered to be very interesting but the article requires an in-depth review of the results and, in particular, a further discussion of childhood asthma in the USA. Also, the introduction, materials and methods are not detailed enough.

Response: Thank you for your constructive comment. We have deepened the analysis of the study results in the “Discussion” section, particularly by adding a discussion of the disease burden of childhood asthma in American children. Our manuscript has been revised as follows: 

“Overall, the prevalence and DALYs of childhood asthma in the USA showed an increasing trend from 1990 to 2019. Part of the reason for this is related to the significant increase in the consumption of sugary drinks such as nutritional/energy drinks, juice drinks, and sweet tea among American children, and sugary drinks are thought to be associated with childhood asthma [1]. However, the upward trend of both indicators from around 2010 to 2019 was significantly reduced compared to that in the period from 2005 to 2009. The prevalence and DALYs of childhood asthma decreased in children aged 1-4 years between 2010 and 2019. This may be related to the enactment of the Clean Air Act amendments in 2011 and the Clean Power Plan in 2015 in the USA. In particular, the Regional Greenhouse Gas Initiative (RGGI) has contributed significantly to the reduction of greenhouse gases in the power sector and toxic air pollutants associated with the onset of asthma, such as PM2.5, over the past decade [2]. Another possible reason is that most US states enacted a tax on sugar-sweetened beverages (SSB) by 2010, which plays a role in reducing SSB-induced high BMI [3]. The prevalence and DALYs of asthma were higher in children in the USA than in those in China. The main reason may be related to the health hypothesis, indicating that less exposure to infection during childhood may result in a greater chance of developing asthma later in life [4]. YLLs in children in both countries account for a minimal proportion of DALYs, and are associated with a very low mortality rate in children with asthma [5]. 

High BMI is a risk factor of great concern for childhood asthma [6]. Our study found an increasing trend in DALYs due to high BMI and the ratio of DALYs rates to DALYs rates due to high BMI in Chinese and American children with asthma. One reason for this is that certain parents of children with asthma are concerned about exercise-induced bronchoconstriction (EIB), and thus impose restrictions on the physical activities of their children, consequently affecting weight management [7]. However, in reality, exercise can increase lung function, promote cardiopulmonary fitness, and control asthma [8]. Adequate warm-up before exercise is also recommended [9]. Another reason is that children with high BMI are more inclined to consume a high-fat diet, which can increase bronchial hyperresponsiveness and exacerbate the symptoms of asthma [10]. Positive effects of weight loss on asthma-related outcomes have been demonstrated [11]. The latest Global Initiative for Asthma (GINA 2021) lists obesity as a modifiable risk factor [6]. Since the global prevalence of high BMI is constantly rising, increasing exercise and reducing high-fat food intake to control high BMI are necessary to reduce the risk of asthma [12]. Although the DALYs rates due to high BMI and the ratio of DALYs rates to DALYs rates due to high BMI were both higher in American children than in Chinese children, the increasing trend of both indicators was significantly higher in Chinese children of almost all ages than in American children. Chinese boys of all ages had a higher percentage of high BMI than girls, which is consistent with the findings of Guo et al. [13]. A study of disease burden, injury, and risk factors by state in the USA from 1990 to 2016 found that high BMI was the most important risk factor in the USA, and that exposure was steadily increasing [14]. They believe that renewed efforts to control weight at the community level are important, and that controlling high BMI needs to be a priority for all stakeholders such as physicians, nurses, policy makers, patients, and families. Liu et al. found that the American government partially eliminated the adverse effects of obesity on asthma by imposing a high-calorie tax, increasing the proportion of nutritious food advertisements, banning the sale of soft drinks, increasing opportunities and venues for physical activity, and implementing better health care policies [15]. Therefore, it is recommended that high BMI be taken more into account in the future development of policies for the prevention, control, and treatment of childhood asthma. Moreover, we recommend that children reduce their BMI by increasing physical activity and eating a healthy diet, which parents should encourage and safeguard [16]. 

The prevalence, DALYs and DALYs due to high BMI were higher in boys than in girls across all age groups in both countries. Boys also have a relatively narrow airway, and are more inclined to vigorous exercise with their greater range of motion, and thus, are more likely to get exposed to allergens. This corroborates the findings of Ellie et al. that boys are more likely to develop allergic diseases than girls based on blood-specific IgE assays and skin prick tests for common allergens [17]. In both China and the USA, the prevalence and DALYs were highest in the 5-9 years age group. This could be due to the fact that children in this age group are at high risk of upper respiratory tract infections because of their low self-management skills and immune levels [13]. In addition, upper respiratory tract infections are important triggers for asthma in children. Moreover, the YLLs for girls aged 10-14 years in the USA have the highest levels than that for girls in the other groups, which is considered to be associated with the increased levels of estrogen and progesterone during the luteal phase in girls of this age group, resulting in increased inflammation of the airway wall [18].” (Page 22, Line 249-261; Page 23, Line 262-283; Page 24, Line 284-305; Page 25, Line 306-322)

In addition, we have expanded the “Introduction” and the “Materials and methods” sections. 

Modifications in the “Introduction” section:

“Asthma is one of the most common chronic diseases in children, with wheezing, coughing, and airflow restriction as clinical manifestations, affecting children's daily life [19]. The incidence, prevalence, and medical costs of this disease have been increasing in recent years [4, 20]. A survey revealed that the prevalence of asthma in Chinese children increased from 0.91% to 2.12% between 1990 and 2010 [21]. Respiratory health during early life may have a lifelong impact on lung health and life expectancy; thus, prevention and control of childhood asthma is particularly crucial to promote individual health and reduce the societal burden of the disease [22]. 

However, the etiology of childhood asthma is yet to be elucidated. Therefore, identifying its risk factors and exploring possible mechanisms is necessary for early detection and intervention to prevent further adverse outcomes [23]. Currently, reported risk factors for asthma include genetic factors, tobacco exposure, dampness/humidity, animal contact, climate, and inhalation of small particles [17, 23, 24]. High body mass index (BMI), which is considered as the seventh-leading level 2 risk factor for attributable disability-adjusted life years (DALYs) of diseases in 2019, is also a risk factor for asthma [25]. It is thought to be associated with dietary habits, lifestyle, and food intake [26].

There are differences in the prevalence and disease burden of asthma between developing and developed countries [4]. The direct and indirect economic costs of childhood asthma are high, and there is a link between the disease burden of asthma and the economic level of the country [27]. It is well known that China is the largest developing country in the world and that the United States of America (USA) is the major developed country [28, 29]. However, to the best of our knowledge, there has been no comparative analysis between China and the USA in these areas. 

Thus, this study aimed to investigate the prevalence of asthma, DALYs, and the effect of the risk factor high BMI on disease burden in children aged 1-14 years in China and the USA, to compare and analyze the differences between them, to provide information for resource allocation, and to learn from the prevention and control strategies of developed countries such as the USA, which can provide some prevention and control strategies to reduce the disease burden of childhood asthma in developing countries such as China.” (Page 4, Line 48-68; Page 5, Line 69-85)

Modifications in the “Materials and methods” section:

“The GBD estimation process uses 86,249 sources that are broad and representative, including censuses, household surveys, health service use, civil registration and vital statistics, air pollution testing, etc.” (Page 6, Line 97-100)

“In this study, we obtained data on the prevalence, DALYs, YLDs, YLLs, and DALYs due to high BMI of childhood asthma in children aged 1-14 years in China and the USA from GBD 2019.” (Page 6, Line 106-108)

“All epidemiological data obtained were age-standardized to match the characteristics of the different national reference populations and finally expressed in terms of 100,000 population [30].” (Page 6, Line 112; Page 7, Line 113-114)

“Comparing AAPC with 0, the curve shows an increasing or decreasing trend with 95% CI not including 0 when the AAPC value is positive or negative. When the 95% CI of AAPC includes 0, the value is stable.” (Page 7, Line 130-132)

Reviewer 2

General Comment

The authors have chosen an interesting study topic to quantify the disease burden due to high body mass index in childhood asthma. The GDB methodology is currently a very important approach to global descriptive epidemiology with the intention of knowledge translation. Hence, the results of the GBD studies should be understandable not only to the academic audience but also to the stakeholders and decision-makers in the health policy domain. In this sense, all improvements to the text should enable the achievement of these goals. There are no ethical issues concerning this paper.

Response: Thank you for this comment. We have added a more in-depth analysis of the discussion, particularly on the disease burden of asthma in children in the USA. In addition, we have added a description of American policies to prevent and control childhood asthma and provided relevant recommendations for policy development in the field of public health and for children's own measures to prevent the disease. Our manuscript has been revised as follows: 

“Overall, the prevalence and DALYs of childhood asthma in the USA showed an increasing trend from 1990 to 2019. Part of the reason for this is related to the significant increase in the consumption of sugary drinks such as nutritional/energy drinks, juice drinks, and sweet tea among American children, and sugary drinks are thought to be associated with childhood asthma [1]. However, the upward trend of both indicators from around 2010 to 2019 was significantly reduced compared to that in the period from 2005 to 2009. The prevalence and DALYs of childhood asthma decreased in children aged 1-4 years between 2010 and 2019. This may be related to the enactment of the Clean Air Act amendments in 2011 and the Clean Power Plan in 2015 in the USA. In particular, the Regional Greenhouse Gas Initiative (RGGI) has contributed significantly to the reduction of greenhouse gases in the power sector and toxic air pollutants associated with the onset of asthma, such as PM2.5, over the past decade [2]. Another possible reason is that most US states enacted a tax on sugar-sweetened beverages (SSB) by 2010, which plays a role in reducing SSB-induced high BMI [3]. The prevalence and DALYs of asthma were higher in children in the USA than in those in China. The main reason may be related to the health hypothesis, indicating that less exposure to infection during childhood may result in a greater chance of developing asthma later in life [4]. YLLs in children in both countries account for a minimal proportion of DALYs, and are associated with a very low mortality rate in children with asthma [5]. 

High BMI is a risk factor of great concern for childhood asthma [6]. Our study found an increasing trend in DALYs due to high BMI and the ratio of DALYs rates to DALYs rates due to high BMI in Chinese and American children with asthma. One reason for this is that certain parents of children with asthma are concerned about exercise-induced bronchoconstriction (EIB), and thus impose restrictions on the physical activities of their children, consequently affecting weight management [7]. However, in reality, exercise can increase lung function, promote cardiopulmonary fitness, and control asthma [8]. Adequate warm-up before exercise is also recommended [9]. Another reason is that children with high BMI are more inclined to consume a high-fat diet, which can increase bronchial hyperresponsiveness and exacerbate the symptoms of asthma [10]. Positive effects of weight loss on asthma-related outcomes have been demonstrated [11]. The latest Global Initiative for Asthma (GINA 2021) lists obesity as a modifiable risk factor [6]. Since the global prevalence of high BMI is constantly rising, increasing exercise and reducing high-fat food intake to control high BMI are necessary to reduce the risk of asthma [12]. Although the DALYs rates due to high BMI and the ratio of DALYs rates to DALYs rates due to high BMI were both higher in American children than in Chinese children, the increasing trend of both indicators was significantly higher in Chinese children of almost all ages than in American children. Chinese boys of all ages had a higher percentage of high BMI than girls, which is consistent with the findings of Guo et al. [13]. A study of disease burden, injury, and risk factors by state in the USA from 1990 to 2016 found that high BMI was the most important risk factor in the USA, and that exposure was steadily increasing [14]. They believe that renewed efforts to control weight at the community level are important, and that controlling high BMI needs to be a priority for all stakeholders such as physicians, nurses, policy makers, patients, and families. Liu et al. found that the American government partially eliminated the adverse effects of obesity on asthma by imposing a high-calorie tax, increasing the proportion of nutritious food advertisements, banning the sale of soft drinks, increasing opportunities and venues for physical activity, and implementing better health care policies [15]. Therefore, it is recommended that high BMI be taken more into account in the future development of policies for the prevention, control, and treatment of childhood asthma. Moreover, we recommend that children reduce their BMI by increasing physical activity and eating a healthy diet, which parents should encourage and safeguard [16]. 

The prevalence, DALYs and DALYs due to high BMI were higher in boys than in girls across all age groups in both countries. Boys also have a relatively narrow airway, and are more inclined to vigorous exercise with their greater range of motion, and thus, are more likely to get exposed to allergens. This corroborates the findings of Ellie et al. that boys are more likely to develop allergic diseases than girls based on blood-specific IgE assays and skin prick tests for common allergens [17]. In both China and the USA, the prevalence and DALYs were highest in the 5-9 years age group. This could be due to the fact that children in this age group are at high risk of upper respiratory tract infections because of their low self-management skills and immune levels [13]. In addition, upper respiratory tract infections are important triggers for asthma in children. Moreover, the YLLs for girls aged 10-14 years in the USA have the highest levels than that for girls in the other groups, which is considered to be associated with the increased levels of estrogen and progesterone during the luteal phase in girls of this age group, resulting in increased inflammation of the airway wall [18].” (Page 22, Line 249-261; Page 23, Line 262-283; Page 24, Line 284-305; Page 25, Line 306-322)

Meanwhile, we have rewritten the conclusions as follows:

“DALYs rates due to high BMI and ratio of DALYs rates to DALYs rates due to high BMI were on the rise in children with asthma in both China and USA. High BMI needs to be taken more into account in the development of future policies for the prevention, control, and treatment of childhood asthma. Although both indicators of asthma in children in the USA are higher than those in China, the increasing trend is significantly lower than that in Chinese children of almost all ages. Therefore, it is recommended to learn from the American government to impose a high-calorie tax, increase physical exercise facilities, and provide better health care policies. Besides, we appeal that children increase their physical activity and maintain a healthy diet and that parents encourage and safeguard it. Children aged 5-9 years had the highest prevalence and DALYs in both countries. Careful attention and targeted intervention should be considered in this population, who are particularly at high risk for asthma.” (Page 26, Line 334-348)

Specific Comment

1. The text in the Introduction (lines 57-59) needs a reference. Consider excluding Ref 8 and deleting the sentence from P 4, line 60. 

Response: Thank you for this comment. We apologize for the error. We have cited Ref 8 in the original lines 57-59 and deleted the sentence in the original line 60:

“High body mass index (BMI), which is considered as the seventh-leading level 2 risk factor for attributable disability-adjusted life years (DALYs) of diseases in 2019, is also a risk factor for asthma [25].” (Page 4, Line 66-68)

2. Also, the authors should include an additional reference which is recently published on this topic (Jin Liu, Maobo Yuan, Yuqian Chen, Yan Wang, Qingting Wang, Qianqian Zhang, Limin Chai, Danyang Li, Yuanjie Qiu, Huan Chen, Jian Wang, Xinming Xie, Manxiang Li. Global burden of asthma associated with high body mass index from 1990 to 2019, Annals of Allergy, Asthma & Immunology 2022; 129:6, 720-730)

Response: Thank you for this comment. We have studied this in depth and cited this reference in our discussion. The revisions are as follows:

“Liu et al. found that the American government partially eliminated the adverse effects of obesity on asthma by imposing a high-calorie tax, increasing the proportion of nutritious food advertisements, banning the sale of soft drinks, increasing opportunities and venues for physical activity, and implementing better health care policies [15].” (Page 24, Line 294-297)

3. The Protocol for the global burden of diseases, injuries, and risk factors study (Institute for Health Metrics and Evaluation, 2018) should be the reference added (and used) to the Method section.

Response: Thank you for this comment. We have added a relevant reference for the GBD protocol to increase the introduction to the GBD database: 

“The Global Burden of Disease (GBD) 2019 is a cross-border collaborative project covering 204 countries and regions. It collected data from disease surveillance sites, surveys of the National Health Service, and published literature data to estimate descriptive epidemiological information on the incidence, prevalence, disability-adjusted life years (DALYs), years of lost due to disability (YLDs), and years of life lost (YLLs) for 369 stratified diseases and injuries using the DisMod-MR 2.1 as a Bayesian meta-regression model [30, 31].” (Page 6, Line 92-97)

4. The results are extensive. However, it seems difficult for authors to extract a meaningful message from the results section to emphasize the rationale for comparing the findings from China and USA. They haven’t concluded anything about disease burden due to the high body mass index in childhood asthma in the USA (P 29, lines 261-269). Thus, I suggest rewriting of the conclusion section.

Response: Thank you for this comment. In the “Discussion” section, we have further analyzed meaningful results and added a discussion of the disease burden of childhood asthma in American children, particularly that resulting from high BMI. Our manuscript has been revised as follows: 

“Overall, the prevalence and DALYs of childhood asthma in the USA showed an increasing trend from 1990 to 2019. Part of the reason for this is related to the significant increase in the consumption of sugary drinks such as nutritional/energy drinks, juice drinks, and sweet tea among American children, and sugary drinks are thought to be associated with childhood asthma [1]. However, the upward trend of both indicators from around 2010 to 2019 was significantly reduced compared to that in the period from 2005 to 2009. The prevalence and DALYs of childhood asthma decreased in children aged 1-4 years between 2010 and 2019. This may be related to the enactment of the Clean Air Act amendments in 2011 and the Clean Power Plan in 2015 in the USA. In particular, the Regional Greenhouse Gas Initiative (RGGI) has contributed significantly to the reduction of greenhouse gases in the power sector and toxic air pollutants associated with the onset of asthma, such as PM2.5, over the past decade [2]. Another possible reason is that most US states enacted a tax on sugar-sweetened beverages (SSB) by 2010, which plays a role in reducing SSB-induced high BMI [3]. The prevalence and DALYs of asthma were higher in children in the USA than in those in China. The main reason may be related to the health hypothesis, indicating that less exposure to infection during childhood may result in a greater chance of developing asthma later in life [4]. YLLs in children in both countries account for a minimal proportion of DALYs, and are associated with a very low mortality rate in children with asthma [5]. 

High BMI is a risk factor of great concern for childhood asthma [6]. Our study found an increasing trend in DALYs due to high BMI and the ratio of DALYs rates to DALYs rates due to high BMI in Chinese and American children with asthma. One reason for this is that certain parents of children with asthma are concerned about exercise-induced bronchoconstriction (EIB), and thus impose restrictions on the physical activities of their children, consequently affecting weight management [7]. However, in reality, exercise can increase lung function, promote cardiopulmonary fitness, and control asthma [8]. Adequate warm-up before exercise is also recommended [9]. Another reason is that children with high BMI are more inclined to consume a high-fat diet, which can increase bronchial hyperresponsiveness and exacerbate the symptoms of asthma [10]. Positive effects of weight loss on asthma-related outcomes have been demonstrated [11]. The latest Global Initiative for Asthma (GINA 2021) lists obesity as a modifiable risk factor [6]. Since the global prevalence of high BMI is constantly rising, increasing exercise and reducing high-fat food intake to control high BMI are necessary to reduce the risk of asthma [12]. Although the DALYs rates due to high BMI and the ratio of DALYs rates to DALYs rates due to high BMI were both higher in American children than in Chinese children, the increasing trend of both indicators was significantly higher in Chinese children of almost all ages than in American children. Chinese boys of all ages had a higher percentage of high BMI than girls, which is consistent with the findings of Guo et al. [13]. A study of disease burden, injury, and risk factors by state in the USA from 1990 to 2016 found that high BMI was the most important risk factor in the USA, and that exposure was steadily increasing [14]. They believe that renewed efforts to control weight at the community level are important, and that controlling high BMI needs to be a priority for all stakeholders such as physicians, nurses, policy makers, patients, and families. Liu et al. found that the American government partially eliminated the adverse effects of obesity on asthma by imposing a high-calorie tax, increasing the proportion of nutritious food advertisements, banning the sale of soft drinks, increasing opportunities and venues for physical activity, and implementing better health care policies [15]. Therefore, it is recommended that high BMI be taken more into account in the future development of policies for the prevention, control, and treatment of childhood asthma. Moreover, we recommend that children reduce their BMI by increasing physical activity and eating a healthy diet, which parents should encourage and safeguard [16]. 

The prevalence, DALYs and DALYs due to high BMI were higher in boys than in girls across all age groups in both countries. Boys also have a relatively narrow airway, and are more inclined to vigorous exercise with their greater range of motion, and thus, are more likely to get exposed to allergens. This corroborates the findings of Ellie et al. that boys are more likely to develop allergic diseases than girls based on blood-specific IgE assays and skin prick tests for common allergens [17]. In both China and the USA, the prevalence and DALYs were highest in the 5-9 years age group. This could be due to the fact that children in this age group are at high risk of upper respiratory tract infections because of their low self-management skills and immune levels [13]. In addition, upper respiratory tract infections are important triggers for asthma in children. Moreover, the YLLs for girls aged 10-14 years in the USA have the highest levels than that for girls in the other groups, which is considered to be associated with the increased levels of estrogen and progesterone during the luteal phase in girls of this age group, resulting in increased inflammation of the airway wall [18].” (Page 22, Line 249-261; Page 23, Line 262-283; Page 24, Line 284-305; Page 25, Line 306-322)

Meanwhile, we have rewritten the conclusions as follows:

“DALYs rates due to high BMI and ratio of DALYs rates to DALYs rates due to high BMI were on the rise in children with asthma in both China and USA. High BMI needs to be taken more into account in the development of future policies for the prevention, control, and treatment of childhood asthma. Although both indicators of asthma in children in the USA are higher than those in China, the increasing trend is significantly lower than that in Chinese children of almost all ages. Therefore, it is recommended to learn from the American government to impose a high-calorie tax, increase physical exercise facilities, and provide better health care policies. Besides, we appeal that children increase their physical activity and maintain a healthy diet and that parents encourage and safeguard it. Children aged 5-9 years had the highest prevalence and DALYs in both countries. Careful attention and targeted intervention should be considered in this population, who are particularly at high risk for asthma.” (Page 26, Line 334-348)

5. In the Discussion section, I suggest that authors delete the sentence (P 22, lines 237-239); it is irrelevant to this paper.

Response: Thank you for your comment. We have deleted the relevant sentence. 

6. I am confused with the text in the Reference section (lines 360-457). Please, check if it is an addition of ref. 22, and, is it necessary.

Response: Thank you for your comment. We apologize for the mistake of Ref. 22. We have checked the style of the cited references and made corrections as follows: 

“31.Reddel HK, Bacharier LB, Bateman ED, Brightling CE, Brusselle GG, Buhl R, et al. Global Initiative for Asthma Strategy 2021: executive summary and rationale for key changes. Eur Respir J. 2022;59(1). doi: 10.1183/13993003.02730-2021.” (Page 33, Line 445-447)

References

1. Xie L, Atem F, Gelfand A, Delclos G, Messiah SE. Association between asthma and sugar-sweetened beverage consumption in the United States pediatric population. J Asthma. 2022;59(5):926-33. doi: 10.1080/02770903.2021.1895210.

2. Perera F, Cooley D, Berberian A, Mills D, Kinney P. Co-Benefits to Children's Health of the U.S. Regional Greenhouse Gas Initiative. Environ Health Perspect. 2020;128(7):77006. doi: 10.1289/EHP6706.

3. Chriqui JF, Eidson S, Chaloupka F. State sales taxes on regular soda (as of January 2014)–BTG Fact Sheet. Bridging the Gap Programme. 2014.

4. Ferrante G, La Grutta S. The Burden of Pediatric Asthma. Front Pediatr. 2018;6:186. doi: 10.3389/fped.2018.00186.

5. Ughasoro MD, Eze JN, Oguonu T, Onwujekwe EO. Burden of childhood and adolescence asthma in Nigeria: Disability adjusted life years. Paediatr Respir Rev. 2021. doi: 10.1016/j.prrv.2021.07.004.

6. Reddel HK, Bacharier LB, Bateman ED, Brightling CE, Brusselle GG, Buhl R, et al. Global Initiative for Asthma Strategy 2021: executive summary and rationale for key changes. Eur Respir J. 2022;59(1). doi: 10.1183/13993003.02730-2021.

7. Kornblit A, Cain A, Bauman LJ, Brown NM, Reznik M. Parental Perspectives of Barriers to Physical Activity in Urban Schoolchildren With Asthma. Acad Pediatr. 2018;18(3):310-6. doi: 10.1016/j.acap.2017.12.011.

8. Global Strategy for Asthma Management and Prevention. Available from: https://ginasthma.org/gina-reports/ (accessed on 1 May 2020).

9. Parsons JP, Hallstrand TS, Mastronarde JG, Kaminsky DA, Rundell KW, Hull JH, et al. An official American Thoracic Society clinical practice guideline: exercise-induced bronchoconstriction. Am J Respir Crit Care Med. 2013;187(9):1016-27. doi: 10.1164/rccm.201303-0437ST.

10. Wood LG. Diet, Obesity, and Asthma. Ann Am Thorac Soc. 2017;14(Supplement_5):S332-S8. doi: 10.1513/AnnalsATS.201702-124AW.

11. Juel CT, Ali Z, Nilas L, Ulrik CS. Asthma and obesity: does weight loss improve asthma control? a systematic review. J Asthma Allergy. 2012;5:21-6. doi: 10.2147/JAA.S32232.

12. Li X, Cao X, Guo M, Xie M, Liu X. Trends and risk factors of mortality and disability adjusted life years for chronic respiratory diseases from 1990 to 2017: systematic analysis for the Global Burden of Disease Study 2017. BMJ. 2020;368:m234. doi: 10.1136/bmj.m234.

13. Guo X, Li Z, Ling W, Long J, Su C, Li J, et al. Epidemiology of childhood asthma in mainland China (1988-2014): A meta-analysis. Allergy Asthma Proc. 2018;39(3):15-29. doi: 10.2500/aap.2018.39.4131.

14. Collaborators USBoD, Mokdad AH, Ballestros K, Echko M, Glenn S, Olsen HE, et al. The State of US Health, 1990-2016: Burden of Diseases, Injuries, and Risk Factors Among US States. JAMA. 2018;319(14):1444-72. doi: 10.1001/jama.2018.0158.

15. Liu J, Yuan M, Chen Y, Wang Y, Wang Q, Zhang Q, et al. Global burden of asthma associated with high body mass index from 1990 to 2019. Ann Allergy Asthma Immunol. 2022;129(6):720-30 e8. doi: 10.1016/j.anai.2022.08.013.

16. Zhang D, Zheng J. The Burden of Childhood Asthma by Age Group, 1990-2019: A Systematic Analysis of Global Burden of Disease 2019 Data. Front Pediatr. 2022;10:823399. doi: 10.3389/fped.2022.823399.

17. Ellie AS, Sun Y, Hou J, Wang P, Zhang Q, Sundell J. Prevalence of Childhood Asthma and Allergies and Their Associations with Perinatal Exposure to Home Environmental Factors: A Cross-Sectional Study in Tianjin, China. Int J Environ Res Public Health. 2021;18(8). doi: 10.3390/ijerph18084131.

18. Melgert BN, Ray A, Hylkema MN, Timens W, Postma DS. Are there reasons why adult asthma is more common in females? Curr Allergy Asthma Rep. 2007;7(2):143-50. doi: 10.1007/s11882-007-0012-4.

19. von Mutius E, Smits HH. Primary prevention of asthma: from risk and protective factors to targeted strategies for prevention. Lancet. 2020;396(10254):854-66. doi: 10.1016/S0140-6736(20)31861-4.

20. Kim A, Lim G, Oh I, Kim Y, Lee T, Lee J. Perinatal factors and the development of childhood asthma. Ann Allergy Asthma Immunol. 2018;120(3):292-9. doi: 10.1016/j.anai.2017.12.009.

21. Li X, Song P, Zhu Y, Lei H, Chan KY, Campbell H, et al. The disease burden of childhood asthma in China: a systematic review and meta-analysis. J Glob Health. 2020;10(1):010801. doi: 10.7189/jogh.10.01081.

22. Postma DS, Bush A, van den Berge M. Risk factors and early origins of chronic obstructive pulmonary disease. Lancet. 2015;385(9971):899-909. doi: 10.1016/S0140-6736(14)60446-3.

23. Yu H, Su F, Wang LB, Hemminki K, Dharmage SC, Bowatte G, et al. The Asthma Family Tree: Evaluating Associations Between Childhood, Parental, and Grandparental Asthma in Seven Chinese Cities. Front Pediatr. 2021;9:720273. doi: 10.3389/fped.2021.720273.

24. Yang X, Zhang Y, Zhan X, Xu X, Li S, Xu X, et al. Particulate matter exposure is highly correlated to pediatric asthma exacerbation. Aging (Albany NY). 2021;13(13):17818-29. doi: 10.18632/aging.203281.

25. Sinyor B, Concepcion Perez L. Pathophysiology Of Asthma. StatPearls. Treasure Island (FL)2022.

26. Kunaratnam K, Halaki M, Wen LM, Baur LA, Flood VM. Tracking Preschoolers' Lifestyle Behaviors and Testing Maternal Sociodemographics and BMI in Predicting Child Obesity Risk. J Nutr. 2020;150(12):3068-74. doi: 10.1093/jn/nxaa292.

27. Soares LON, Theodoro EE, Angelelli MM, Lin LL, Carchedi GR, Silva CC, et al. Evaluating the effect of childhood and adolescence asthma on the household economy. J Pediatr (Rio J). 2022;98(5):490-5. doi: 10.1016/j.jped.2021.12.010.

28. Qiu H, Cao S, Xu R. Cancer incidence, mortality, and burden in China: a time-trend analysis and comparison with the United States and United Kingdom based on the global epidemiological data released in 2020. Cancer Commun (Lond). 2021;41(10):1037-48. doi: 10.1002/cac2.12197.

29. Wen H, Xie C, Shi F, Liu Y, Liu X, Yu C. Trends in Deaths Attributable to Smoking in China, Japan, United Kingdom, and United States From 1990 to 2019. Int J Public Health. 2022;67:1605147. doi: 10.3389/ijph.2022.1605147.

30. Diseases GBD, Injuries C. Global burden of 369 diseases and injuries in 204 countries and territories, 1990-2019: a systematic analysis for the Global Burden of Disease Study 2019. Lancet. 2020;396(10258):1204-22. doi: 10.1016/S0140-6736(20)30925-9.

31. Metrics IfH, Evaluation. Protocol for the Global Burden of Diseases, Injuries, and Risk Factors Study (GBD). IHME Seattle, WA; 2015.

---

## [Decision Letter · Decision Letter 1]

13 Mar 2023

Analysis of disease burden due to high body mass index in childhood asthma in China and the USA based on the Global Burden of Disease Study 2019

PONE-D-22-30916R1

Dear Dr. Pan,

We’re pleased to inform you that your manuscript has been judged scientifically suitable for publication and will be formally accepted for publication once it meets all outstanding technical requirements.

Kind regards,

Gorica Maric

Academic Editor

PLOS ONE

Additional Editor Comments (optional):

Reviewers' comments:

Reviewer's Responses to Questions

**Comments to the Author**

1. If the authors have adequately addressed your comments raised in a previous round of review and you feel that this manuscript is now acceptable for publication, you may indicate that here to bypass the “Comments to the Author” section, enter your conflict of interest statement in the “Confidential to Editor” section, and submit your "Accept" recommendation.

Reviewer #1: All comments have been addressed

Reviewer #2: All comments have been addressed

2. Is the manuscript technically sound, and do the data support the conclusions?

Reviewer #1: Partly

Reviewer #2: Yes

3. Has the statistical analysis been performed appropriately and rigorously? 

Reviewer #1: Yes

Reviewer #2: Yes

4. Have the authors made all data underlying the findings in their manuscript fully available?

Reviewer #1: Yes

Reviewer #2: Yes

5. Is the manuscript presented in an intelligible fashion and written in standard English?

Reviewer #1: Yes

Reviewer #2: Yes

6. Review Comments to the Author

Reviewer #1: Dear authors:

Thank you very much for your corrections.

I believe that the results could better support the conclusions. Their presentation is long and can be confusing.

Yours sincerely.

Reviewer #2: Dear authors, thank you for seriously approaching revising your paper. As for my suggestions, they have all been addrested, and the article has been significantly improved. My recommendation to the editor is to accept the paper.

7. PLOS authors have the option to publish the peer review history of their article (what does this mean?). If published, this will include your full peer review and any attached files.

Reviewer #1: **Yes: **Enrique Gea-Izquierdo

Reviewer #2: **Yes: **Prof. dr Bojana Matejić

---

## [Editor Report · Acceptance letter]

17 Mar 2023

PONE-D-22-30916R1 

Analysis of disease burden due to high body mass index in childhood asthma in China and the USA based on the Global Burden of Disease Study 2019 

Dear Dr. Pan:

I'm pleased to inform you that your manuscript has been deemed suitable for publication in PLOS ONE. Congratulations! Your manuscript is now with our production department. 

Kind regards, 

on behalf of

Dr. Gorica Maric 

Academic Editor

PLOS ONE